# Moyamoya disease factor RNF213 is a giant E3 ligase with a dynein-like core and a distinct ubiquitin-transfer mechanism

Juraj Ahel[1], Anita Lehner[2], Antonia Vogel[1], Alexander Schleiffer[1], Anton Meinhart[1], David Haselbach[1]*, Tim Clausen[1,3]*

[1]Research Institute of Molecular Pathology (IMP), Vienna BioCenter, Vienna, Austria; [2]Vienna BioCenter Core Facilities, Vienna BioCenter, Vienna, Austria; [3]Medical University of Vienna, Vienna, Austria

**Abstract** RNF213 is the major susceptibility factor for Moyamoya disease, a progressive cerebrovascular disorder that often leads to brain stroke in adults and children. Characterization of disease-associated mutations has been complicated by the enormous size of RNF213. Here, we present the cryo-EM structure of mouse RNF213. The structure reveals the intricate fold of the 584 kDa protein, comprising an N-terminal stalk, a dynein-like core with six ATPase units, and a multidomain E3 module. Collaboration with UbcH7, a cysteine-reactive E2, points to an unexplored ubiquitin-transfer mechanism that proceeds in a RING-independent manner. Moreover, we show that pathologic MMD mutations cluster in the composite E3 domain, likely interfering with substrate ubiquitination. In conclusion, the structure of RNF213 uncovers a distinct type of an E3 enzyme, highlighting the growing mechanistic diversity in ubiquitination cascades. Our results also provide the molecular framework for investigating the emerging role of RNF213 in lipid metabolism, hypoxia, and angiogenesis.

*For correspondence:
tim.clausen@imp.ac.at (TC);
david.haselbach@IMP.AC.AT (DH)

**Competing interests:** The authors declare that no competing interests exist.

## Introduction

RNF213, also known as 'mysterin', is the major susceptibility gene for Moyamoya disease (MMD), a cerebrovascular disorder characterized by arterial occlusions and abnormal blood vessel generation (*Liu et al., 2011*; *Lutterman et al., 1998*; *Morito et al., 2015*). MMD is most prevalent in East Asia, where at least 16 million people are carriers of RNF213 mutations, with onset of disease in >50,000 cases (*Liu et al., 2012*). Although the pathologic role of RNF213 is little understood, recent studies highlight its function as a metabolic gatekeeper. It was shown that RNF213 plays an important role in lipid metabolism modulating lipotoxicity (*Piccolis et al., 2019*), fat storage, and lipid droplet formation (*Sugihara et al., 2019*). Moreover, RNF213 together with PTP1B and HIF1A coordinates the cellular response to hypoxia, controlling non-mitochondrial oxygen consumption (*Banh et al., 2016*). Proteomic searches for cellular targets revealed a connection to the ubiquitin-proteasome system, where RNF213 knock-down causes gross changes in the ubiquitome, in particular of members of the NF-κB pathway (*Banh et al., 2016*; *Piccolis et al., 2019*). In fact, RNF213 is capable of activating NF-κB signaling and could thus, by inducing the expression of inflammatory cytokines such as interleukin-6, influence angiogenesis and MMD (*Piccolis et al., 2019*). Still, its exact role in these diverse biological processes needs to be further investigated. The major obstacle in understanding RNF213 is the lack of structural and biochemical data, hampered by the enormous size and complexity of the protein. Notably, with a mass of 591 kDa, RNF213 is the largest E3 ubiquitin ligase in the human proteome. It combines RING and AAA (ATPase associated with a variety of cellular activities) domains in a single polypeptide, making it a unique E3 machine in the ubiquitination system. In the present study, we applied an integrative biochemical and structural approach to reconstitute

**eLife digest** Moyamoya disease is a genetic disorder affecting both adults and children. It is characterized by narrowing of the blood vessels in the brain, which can lead to strokes. Moyamoya patients often have mutations in the gene for a protein called RNF213. This protein is linked to multiple processes in the body, including the development of blood vessels. Despite this, its role in Moyamoya disease is still something of a mystery.

RNF213 is known to fall into two protein 'classes'. First, it is an E3 enzyme. This type of protein tags unwanted or defective proteins for disposal by the cell. Second, it is a motor protein. Motor proteins contain tiny molecular 'engines', called ATPases, that normally convert chemical energy to movement. No other human protein combines these two activities, making RNF213 unique.

RNF213 is also an extremely large protein, which means it is difficult to manipulate in the laboratory and thus hard to study. Scientists still need more detailed information on RNF213's structure and chemical activity before we can understand what the mutant protein might be doing in Moyamoya disease. Ahel et al. therefore set out to make the RNF213 protein and 'dissect' it in a test tube.

Electron microscopy experiments using the mouse-version of RNF213 revealed that it consisted of a single, giant molecule, folded up to form three regions with distinct structures. These were a long 'arm' at one end, a ring-shaped part in the middle, containing the ATPase 'motor', and the E3 enzyme module at the other end.

Further chemical analysis showed that RNF213's ATPase and E3 modules worked in unexpected ways. Although the ATPase did resemble another well-known motor protein, in RNF213 it did not generate movement but rather appeared to act like an intricate molecular 'switch'. The E3 module of RNF213 'tagged' other molecules as expected but did not contain an additional structure that all other known E3 enzymes need to work properly. This suggests that RNF213 represents a distinct class of E3 enzymes.

Biochemical tests of the mutation most commonly found in Moyamoya patients revealed that it left RNF213's overall structure, ATPase motor and E3 module intact. That is, the disease-causing mutation appeared to hinder interactions with other partner proteins, rather than disrupting RNF213 itself.

By providing the first detailed molecular description of the architecture of RNF213, Ahel et al. hope that these findings will help future investigations of both this giant protein's biological role in the cell and its contribution to Moyamoya disease.

---

RNF213, address its structure and mechanism, and analyze the molecular basis of its MMD-causing mutations.

## Results

### Cryo-EM structure of RNF213

RNF213 is a highly conserved protein expressed in all vertebrates (*Figure 1a*, *Supplementary file 1*). Despite the lack of annotated domains for >4000 residues, the giant E3 ligase is predicted to be a structured protein with few disordered regions (*Figure 1—figure supplement 1a*). Consistent with this, the full-length mouse RNF213 (584 kDa, 5148 residues) analyzed in the present study is a stable protein that could be efficiently produced in insect cells and analyzed by single-particle cryo-EM (*Figure 1—figure supplement 2*). The initial cryo-EM class averages revealed a compact macromolecule with overall dimensions of 90 × 130 × 220 Å³, depicting the RNF213 ubiquitin ligase in its monomeric state. Size exclusion chromatography (SEC) and dynamic light scattering (DLS) analyses pointed to a monodisperse protein population (*Figure 1bc*), suggesting that the monomer is the dominant form of RNF213 in solution. To cope with the conformational flexibility within the RNF213 particle (*Video 1*), we performed focused refinements, masking the densities of the mobile portions and processing them separately (*Figure 1—figure supplement 3*). Although model building was complicated by the lack of available homologous structures, the high quality of the focused cryo-EM maps (*Video 2*) enabled us to build the RNF213 structure de novo. During the whole process, the

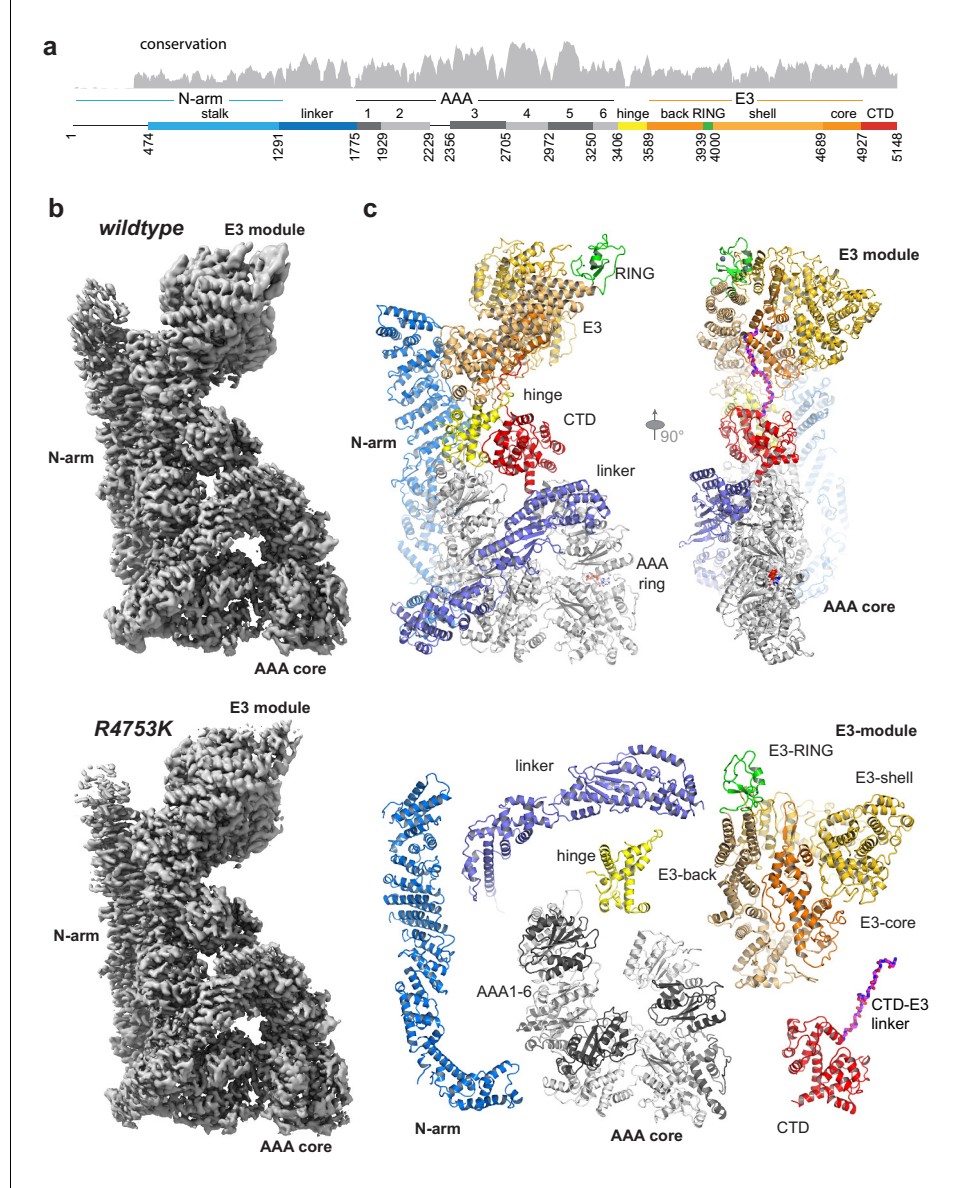

**Figure 1.** Overall structure of the AAA-E3 ligase RNF213. (a) Conservation and domain organization of RNF213. (b) Composite cryo-EM maps of wt and R4753K RNF213 resulting from focused refinements. (c) Architecture of RNF213, shown in orthogonal views (top row; first view matches orientation of panel b), illustrated as ribbon model using a domain-based color mode similar as in (a). The lower panel depicts the collection of dissected RNF213 domains.

The online version of this article includes the following figure supplement(s) for figure 1:

**Figure supplement 1.** In silico and in vitro characterization of RNF213.
**Figure supplement 2.** Cryo-EM analysis of RNF213.
**Figure supplement 3.** Reconstruction of RNF213 cryo-EM densities.
**Figure supplement 4.** XL-MS analysis of RNF213.

correct tracing of the polypeptide chain was carefully validated against cross-linking mass spectrometry (XL-MS) data (*Figure 1—figure supplement 4*). The final atomic models of wildtype (wt) RNF213 and the R4753K MMD mutant comprise residues 476–5148, contain >80% of all side chains and were resolved at an average resolution of 3.2 Å and 3.1 Å, respectively, with the best-refined regions reaching 2.8 Å (*Figure 1—figure supplement 2*, *Table 1*).

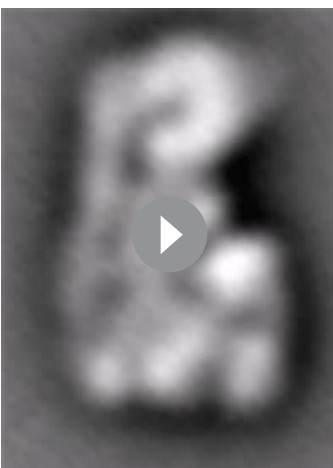

**Video 1.** Dynamics of the RNF213 molecule. The movie shows representative states seen in negative staining EM, in which the E3 module moves relative to the rest of the protein, most noticeably changing the distance between the E3 module and the proximal end of the N-arm.

https://elifesciences.org/articles/56185#video1

Our cryo-EM structure shows that RNF213 is a multipartite AAA-E3 machine. Its intricate protein fold arranges 20 sub-domains in a zig-zag meandering manner into three structural components, to which we refer as N-arm, AAA, and E3 module (*Figure 1*, *Video 3*). The individual components were visualized at an average resolution of 3.4, 3.0, and 3.1 Å, respectively. The N-arm (residues 1–1290) is composed of a disordered region followed by a string of helical bundles that yield a 180 Å-long stalk leaning against the AAA and E3 portions. A linker (1291–1774) connects the N-arm to the second module, the AAA core, which is composed of six non-equivalent AAA units (1775–3405, AAA1 to AAA6). The midpoint of the RNF213 molecule consists of the hinge domain (3406–3588), which connects the AAA core to the third module, harboring the ubiquitin ligase activity. This E3 module (3589–4926) is composed of a heart-shaped, 4-domain scaffold that positions the E3-RING (3940–3999) at the edge of the RNF213 molecule, opposite to the AAA core. At its distal end, the E3 fold embraces a 20-residue loop that returns to the center of the molecule (*Figure 1c*), placing the CTD – a bilobal α-helical domain – at the AAA-E3 interface.

## RNF213 has a dynein-related ATPase core

One of the most prominent features of RNF213 is the presence of a six-membered AAA ring (*Figure 2a*). Of note, only two of the AAA domains had been identified in previous analyses, and given their homology to Hsp100 unfoldases, RNF213 was proposed to form hexameric particles to yield the active AAA ATPase (*Morito et al., 2015*). Contrary to this, the present structure demonstrates that the protein encodes six AAA units within a single polypeptide chain. As such, RNF213 is equivalent to dynein, which was found to be the closest structural homolog of RNF213 (*Figure 2—figure supplement 1*). In general, AAA ATPase domains are composed of a large α/β (L) and a small α-helical (S) domain, with the nucleotide binding site located at their interface (*Erzberger and Berger, 2006*). While the L/S domains carry most motifs for ATP binding (Walker A) and hydrolysis (Walker B, sensor I/II), one critical catalytic residue, the arginine finger (RF), is provided by the L-domain of the adjacent AAA unit, enabling signaling between AAA 'rigid bodies' (*Wang et al., 2001*). Analysis of individual ATPase motifs revealed that in RNF213 only AAA3 and AAA4 are catalytically competent, bearing all functional motifs (*Figure 2b*). Moreover, the cryo-EM density unambiguously revealed a co-purified ATP molecule that is bound to AAA2 (*Figure 2a*, *Video 2*). Consistent with the unexpected density, AAA2 contains a proper nucleotide binding site, but lacks catalytic residues (Walker B, sensor I) required for ATP hydrolysis. Instead, residues from both AAA2 and AAA3 tightly coordinate ATP, making

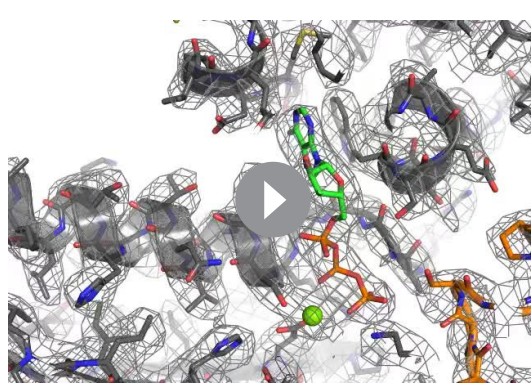

**Video 2.** Cryo-EM density in the central AAA region. The movie illustrates the nucleotide binding site of AAA2 and the regulatory insertions IR5 (orange) and IR3 (orange) located near the AAA4 and AAA3 active sites, respectively. The bound ATP (AAA2) is highlighted in green.

https://elifesciences.org/articles/56185#video2

**Table 1.** Cryo-EM data collection, refinement and validation statistics.

| | Wild type (EMDB-10429) (PDB 6TAX) | R4753K (EMDB-10430) (PDB 6TAY) |
|---|---|---|
| Data collection and processing | | |
| Magnification | 130kx | 130kx |
| Voltage (kV) | 300 | 300 |
| Electron exposure (e–/Å$^2$) | 59.9 | 47.3 |
| Defocus range (μm) | –1.5 to –3.5 | –0.8 to –2.0 |
| Pixel size (Å) | 1.04 | 1.04 |
| Symmetry imposed | C1 | C1 |
| Initial particle images (no.) | | |
| Final particle images (no.) | 374683 | 426312 |
| Map resolution (Å) FSC threshold | 3.2 0.143 | 3.2 0.143 |
| Map resolution range (Å) | 3.1-6.5 | 2.9-6.5 |
| Refinement | | |
| Initial model used (PDB code) | none | none |
| Model resolution (Å) FSC threshold | 3.3 0.5 | 3.3 0.5 |
| Model resolution range (Å) | 3.1-6.5 | 2.9-6.5 |
| Map sharpening B factor (Å$^2$) | –50 | –50 |
| Model composition Non-hydrogen atoms Protein residues Ligands | 35241 4383 ATP, Zn, Mg | 35235 4382 ATP, Zn, Mg |
| B factors (Å$^2$) Protein Ligand (ATP) | 86 64 | 64 50 |
| R.m.s. deviations Bond lengths (Å) Bond angles (°) | 0.011 1.17 | 0.007 1.01 |
| Validation MolProbity score Clashscore Poor rotamers (%) | 1.9 5.9 0.5 | 1.7 4.6 0.2 |
| Ramachandran plot Favored (%) Allowed (%) Disallowed (%) | 90.1 9.8 0.2 | 91.8 8.2 0.1 |

the nucleotide a molecular glue tethering the two domains (*Figure 2—figure supplement 2a*). Furthermore, comparison with canonical AAA hexamers revealed RNF213-specific insertions (*Figure 2ab*), most of which seem to have a structural role stabilizing the rigid bodies within the AAA ring. Curiously, two insertions hint at a regulatory function. First, AAA3L harbors a 51-residue loop (IR3, 2487–2538) that protrudes in a well-defined conformation to the adjacent AAA4. The cryo-EM structure suggests that this RNF213 signature motif functions as a molecular wedge, separating AAA3 and AAA4. Comparison with nucleotide-bound dynein (*Bhabha et al., 2014*) illustrates this effect for the AAA4 arginine finger, which is trapped in a remote position to the AAA3 active site, such that it cannot participate in ATP sensing and hydrolysis (*Figure 2—figure supplement 2b*). Second, AAA5L contains a 40-residue insertion (IR5, 3063–3103) forming an αβ$_3$ structure at the AAA4/AAA5 interface. At the edge of IR5, Tyr3078 reaches over toward the active site of AAA4, sterically blocking nucleotide binding (*Figure 2a*). Together, the two insertions IR3 and IR5, which are amongst the most conserved RNF213 protein stretches (*Figure 2—figure supplement 2c*), should

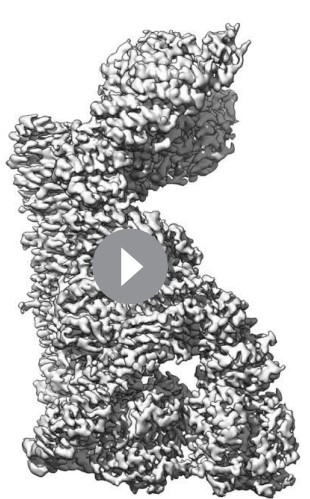

**Video 3.** Overall structure of RNF213. The movie shows the overall architecture of RNF213 by highlighting the different domains from N- to C-terminus and their respective cryo-EM densities.
https://elifesciences.org/articles/56185#video3

hinder nucleotide binding to AAA3 and AAA4, diminishing RNF213 ATPase activity. Both inhibitory loops are tightly bound to the AAA ring, suggesting that their displacement and concomitant activation of the ATPase requires a yet unknown regulatory event. We thus presume that the current cryo-EM structure reflects the auto-inhibited form of the RNF213 ATPase. Our structural data suggest that the latent ATPase state is characterized by an opened AAA ring (*Figure 2ab*). A 20 Å gap between AAA6 and AAA1 divides the hexamer into two halves, AAA1/2/3 and AAA4/5/6, which may reorient en-bloc upon nucleotide-dependent structural changes in AAA3 and AAA4. A similar open-closed transition of the AAA ring has been observed for dynein (*Bhabha et al., 2014*; *Schmidt et al., 2012*). Detailed structural comparisons revealed that RNF213 matches best to the resting states of dynein – the phi-particle (*Zhang et al., 2017*) and the apo form (*Schmidt et al., 2012*; *Figure 2c*, *Figure 2—figure supplement 1*). Overall, the structural alignments highlight the analogous AAA2/3/4 architecture between dynein and RNF213, with functional ATPase units at positions 3 and 4, and an ATP-bound, catalytically inactive AAA2. Moreover, the upstream linker domain adopts a similar fold and position above the RNF213 and dynein AAA rings (*Figure 2c*), despite lacking any sequence similarity, and may therefore regulate ATPase function in an equivalent manner. An important mechanistic difference, however, relates to AAA1. In dynein, AAA1 is the most active ATPase driving processive movement along microtubules (*Kon et al., 2004*; *Reck-Peterson and Vale, 2004*), while AAA3/AAA4 are proposed to function as a molecular switch turning the dynein motor on and off (*Bhabha et al., 2014*). In contrast, AAA1 of RNF213 lacks Walker A and Walker B motifs altogether and should be incapable of binding and hydrolyzing ATP and functioning as processive motor. Consistent with this, we observed that the ATPase activity of RNF213 is an order of magnitude lower than that of a processive AAA machine (*Figure 2d*). Moreover, the basal activity of RNF213 is ~3 fold lower than that of dynein at rest, pointing to an additional inhibitory element controlling AAA activity, as predicted by the cryo-EM structure.

## RNF213 is a distinct E3 ubiquitin ligase

The ubiquitin ligase activity of RNF213 is expected to depend on the E3-RING domain, which is located on top of the E3 scaffold composed by the E3-back, E3-shell, and E3-core domains (*Figure 1c*, *Figure 3a*). The E3-RING, inserted between the two ultimate helices of the E3-back, contains a C4HC3 motif coordinating two $Zn^{2+}$ ions in a cross-braced arrangement (*Metzger et al., 2014*). To delineate its mechanistic features, we aligned E3-RING with the TRIM25/Ubc13 ~ Ub complex (*Sanchez et al., 2016*), the most closely related RING E3 characterized with a ubiquitin-loaded E2 (*Figure 3b*). In RING/E2 ~Ub complexes, ubiquitin is covalently bound via a strained thioester to the E2, facilitating nucleophilic attack of a substrate's lysine residue (*Plechanovová et al., 2012*). Key for this lysine-reactivity is the RING 'linchpin', a conserved Lys/Arg residue that locks the E2 and Ub in the 'closed' conformation primed for catalysis (*Pruneda et al., 2012*). In RNF213, however, the Lys/Arg linchpin is replaced by a Leu (*Figure 3b*), a hydrophobic residue that is unlikely to stabilize the reactive E2 ~Ub intermediate (*Koliopoulos et al., 2016*; *Pruneda et al., 2012*). To characterize the E3 activity in vitro, we followed the auto-ubiquitination of RNF213. To this end, we performed an E2 screen testing 34 ubiquitin-conjugating enzymes and observed that UbcH7 is most efficient in enabling the Ub transfer activity. The stimulated activity was in this case even 2–3 times

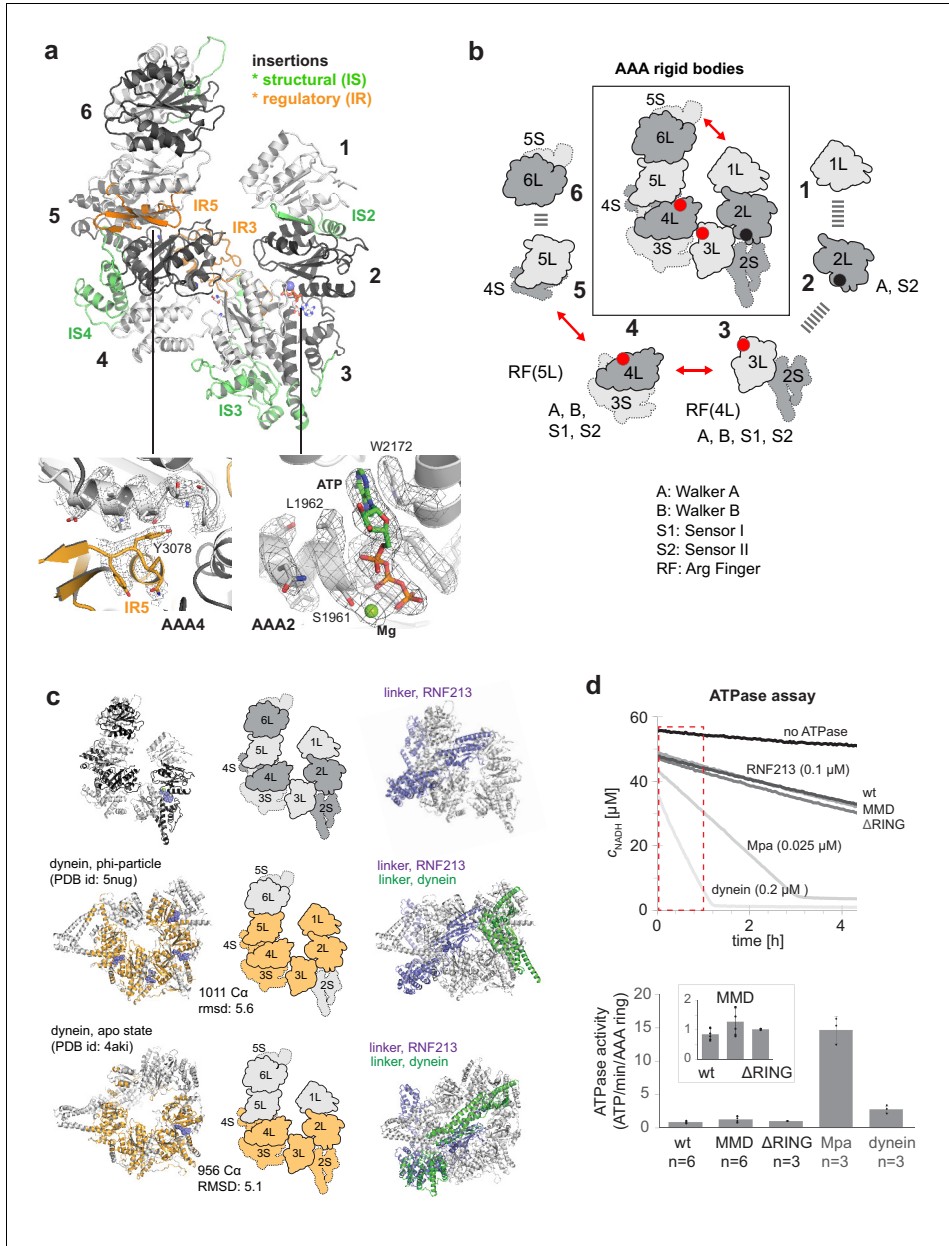

**Figure 2.** AAA core adopts a dynein-like fold. (**a**) Organization of the 6 AAA domains, shown in alternating gray tones, highlighting insertions with structural (IS, green) and regulatory (IR, orange) roles. Orientation is similar as in *Figure 1c*. The insets below illustrate the EM density of the IR5 motif blocking via Tyr3078 nucleotide binding to AAA4 and the ATP molecule tightly bound at AAA2 (also shown in *Video 1*). (**b**) Schematic cartoon of the ATPase core emphasizing the AAA 'rigid bodies' formed between the L domain of one and the S domain of the previous unit (*Wang et al., 2001*). The catalytically competent ATPase sites (red dots), the ATP-bound AAA2 (black dot), and the respective functional motifs for ATP binding and hydrolysis are indicated. (**c**) Structural superposition of RNF213 (top row) with two dynein states (Phi particle, middle; apo dynein, bottom). The left panel illustrates the aligned dynein domains (orange), the middle shows the corresponding cartoon of the matching RNF213 AAA portions, and the right panel highlights the upstream linker of RNF213 (lilac) superimposed with phi-particle and apo dynein structures (AAA core, grey; linker, green). (**d**) ATPase assay comparing RNF213 variants with dynein and a processive ATPase, the unfoldase Mpa. The top panel illustrates representative NADH decay curves, with used protein concentrations being indicated. The region for calculating reaction rates is highlighted in red. The lower panel quantifies the respective ATPase rates calculated per AAA ring. Rates for tested RNF213 variants are also shown enlarged. Error bars indicate standard deviation.

The online version of this article includes the following figure supplement(s) for figure 2:

*Figure 2 continued on next page*

*Figure 2 continued*

**Figure supplement 1.** Structural comparison of RNF213 with dynein-like proteins, containing 6 AAA units in a single polypeptide.
**Figure supplement 2.** Conservation pattern of RNF213 mapped to the cryo-EM structure.
**Figure supplement 3.** Structural comparison of RNF213 and Rea1.

higher than with UbcH5 variants, which are promiscuous E2 enzymes working with many E3 ligases (*Figure 3c*). This finding is surprising, because UbcH7 is not known to collaborate with canonical RING-type E3s as it lacks intrinsic reactivity with lysine. Instead, UbcH7 is reactive only with cysteine residues and works together with transthiolation E3 enzymes such as HECT (*Schwarz et al., 1998*), RBR (*Wenzel et al., 2011*), and RCR (*Pao et al., 2018*) ubiquitin ligases. Support for the here proposed cysteine-dependent ubiquitination activity of RNF213 stems from a recent E3 proteomics analysis (*Pao et al., 2018*). Using an activity-based E2 probe, RNF213 was identified as one of the most prominent E3 ligases collaborating with UbcH7, outranking its established partner enzymes, such as LUBAC, PARKIN, and HERC2. To further corroborate that RNF213 is capable to promote the ubiquitin transfer by a transthiolation mechanism rather than by activating the E2-Ub conjugate, we generated a ΔRING mutant, replacing residues 3941–3999 with a Gly-Ser-Gly-Ser-Gly linker. Strikingly, ΔRING exhibited wt-like auto-ubiquitination activity with either UbcH7 or UbcH5c (*Figure 3d*, *Figure 3—figure supplement 1*), indicating that the RING itself is not involved in the ubiquitin transfer. These data contrast with a previous analysis claiming a critical role of the RING for the ubiquitination reaction (*Liu et al., 2011*). However, that analysis was carried out with a deletion far beyond the RING domain (corresponding to Δ3947–4042 in mouse RNF213) that presumably disrupted the E3 fold, highlighting the value of precise structural information for performing detailed mechanistic analyses. On top of that, a DALI search for structural homologs revealed that the E3 subdomains – namely the E3-back, the E3-shell, the E3-core, and the CTD – do not have counterparts in the PDB database. The active ΔRING mutant therefore shares no similarity to any known E3 ligase either on the sequence or on the structural level, suggesting that RNF213 employs a yet undescribed E3 scaffold to perform its ubiquitin ligase function in a RING-independent manner.

## MMD mutations cluster in the E3 domain

Finally, we investigated the molecular basis of RNF213 mutations causing the MMD. We concentrated our analysis on the human R4810K founder mutation, the most widespread disease variant present in more than 2.0% of the East Asian population. Cryo-EM analysis of the equivalent R4753K mutation in mouse RNF213 indicated that wt and R4753K proteins have an almost identical structure and can be aligned with a root mean square (rms) deviation of 0.69 Å for 4308 Cα atoms. The well-defined EM density at the mutation site allowed us to directly compare the two atomic models and identify structural differences. Arg4753 is located at the end of an α-helix, fastening the ends of an 'omega' structure formed by residues 4750–4809 (*Figure 4a*). Asp4806, the partner residue on the opposite flank of this motif, undergoes close interactions with Arg4753, as directly seen in the cryo-EM density (*Figure 4b*). The R4753K mutation disrupts this salt bridge, weakening the sealing of the omega motif and leading to small but significant structural adaptations in its immediate environment (*Figure 4b*). Interestingly, mutating the Arg4753 partner residue Asp4806 is also associated with MMD (*Liu et al., 2011*), highlighting the relevance of the identified structural motif. To functionally characterize R4753K, we performed ATPase and ubiquitination assays. These data indicated that the MMD founder mutation, though causing local structural changes, does not significantly alter the enzymatic properties of RNF213 in vitro (*Figure 2d*, *Figure 4c*). Consistent with the minor structural and functional effects, the penetrance of R4753K in MMD is low, as only 1 from 200 carriers develops disease (*Liu et al., 2011*). To obtain a comprehensive overview how MMD mutations affect RNF213 function, we mapped known variants on the cryo-EM structure, color coded by their pathology scores from Combined Annotation Dependent Depletion analysis (CADD *Kircher et al., 2014*; *Moteki et al., 2015*; *Figure 4d*, *Figure 4—figure supplement 1*). This plot reveals that MMD mutations strongly cluster in the E3 module, with 21 out of 28 severe RNF213 mutations (CADD-score >20) localized in one of its four E3 sub-domains (*Supplementary file 2*). Remarkably, the mutations with the highest predicted pathogenicity are located at the subdomain interfaces and

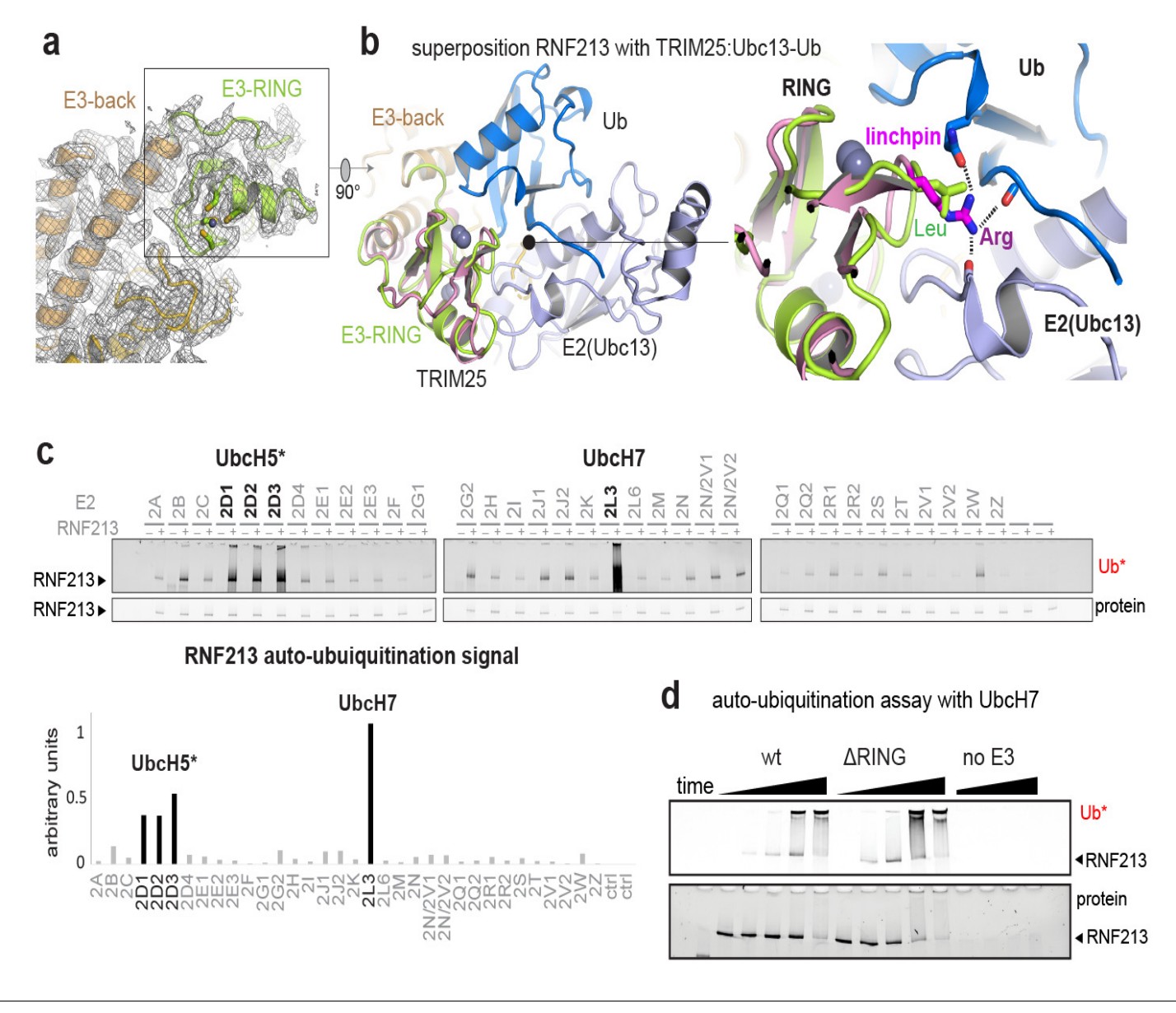

**Figure 3.** E3 ligase function of RNF213. (a) EM density of the RNF213 RING (green ribbon) and its engaging structural elements in the E3 domain. (b) Superposition with the TRIM25/Ubc13 ~Ub complex (PDB id 5eya; colored in brown, slate, blue, respectively), showing the relative orientation of the three domains in the activated 'closed' conformation. $Zn^{2+}$ ions are shown as spheres. The right panel illustrates the catalytic effect of the Arg linchpin in TRIM25, aligning Ub and Ubc13 (important hydrogen bonds are indicated as dashed lines). The corresponding RNF213 linchpin, Leu3986, cannot undergo such interactions. (c) E2 screen, monitoring RNF213 auto-ubiquitination with various E2 enzymes. The panel below shows the quantification of the RNF213-associated poly-Ub signal, revealing the pronounced E3 activity with UbcH7 relative to the indicated UbcH5 variants. (d) Auto-ubiquitination assay comparing the activities of wt and ΔRING RNF213 in the presence of UbcH7. Equivalent results are seen with UbcH5c (*Figure 3—figure supplement 1*).

The online version of this article includes the following figure supplement(s) for figure 3:

**Figure supplement 1.** Auto-ubiquitination activity of RNF213 in the presence of UbcH5c.

should alter the overall conformation and dynamics of the composite E3 module, and thus its ubiquitination activity.

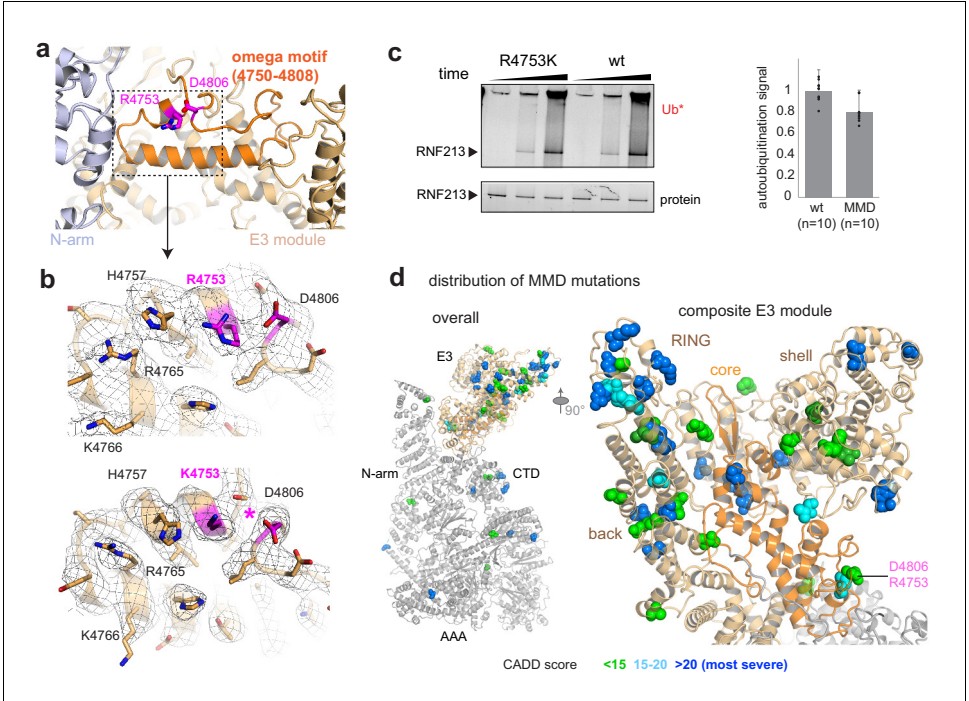

**Figure 4.** Structural analysis of MMD mutations. (**a**) Arg4753 stabilizes the 'omega' motif (orange), located in the periphery of the E3 domain. Arg4753 and its partner residue Asp4806 are highlighted. (**b**) Focused cryo-EM maps of wt (top) and R4753K (bottom), showing that the introduced mutation disrupts the Arg4753-Asp4806 salt bridge (pink), leading to local structural changes. Affected side chains are labelled. (**c**) Auto-ubiquitination assay with UbcH7 comparing the wt and the R4753K mutant. The bar plot represents the quantification of the poly-ubiquitination signal (Ub*), resulting from ten replicated measurements. Error bars indicate standard deviation. (**d**) MMD-associated mutations (colored spheres) mapped on the cryo-EM structure of RNF213 cluster in the E3 domain, though separated in sequence by ~1000 residues. The zoomed-in window (right panel) shows that all four portions of the E3 module (core sub-domain shown in orange) carry MMD mutations. Residues are colored by CADD score (0–15 green, 15–20 cyan, 20+ blue; listed in *Supplementary file 2*). The most pathologic mutations (blue) are individually labeled in *Figure 4—figure supplement 1*.

The online version of this article includes the following figure supplement(s) for figure 4:

**Figure supplement 1.** Cryo-EM structure of RNF213 with labeled MMD mutations.

## Discussion

Moyamoya Disease (MMD) is a severe cerebrovascular disorder leading to early onset of brain stroke in adults and even young children. More than 2% of the East Asian population carry disease-associated mutations in RNF213. Upon addressing its pathologic role in MDD, recent work revealed the involvement RNF213 in hypoxia, lipid metabolism, NF-κB signalling, and angiogenesis (*Banh et al., 2016*; *Piccolis et al., 2019*; *Sugihara et al., 2019*). Moreover, it was shown that RNF213 combines AAA ATPase and E3 activities in a single polypeptide (*Kamada et al., 2011*), yielding a unique ubiquitination machine in the ubiquitin-proteasome system. The current study presents the cryo-EM structure of this giant E3 ligase, depicting the structural organization of the wild-type protein and the MMD founder mutant R4810K. Together with a detailed functional analysis, these data depict a distinctive AAA-E3 enzyme exhibiting various structural and mechanistic peculiarities. As seen in the high-resolution cryo-EM structure, the 584 kDa RNF213 protein adopts an intricately intertwined fold that is organized into three major modules, namely an N-terminal stalk, a dynein-like core with six AAA units, and a multidomain E3 module. The absence of structural counterparts to the E3 module subdomains in the PDB database apart from the RING domain highlights the distinct architecture of the RNF213 ubiquitin ligase.

One of the most remarkable features discovered by the cryo-EM analysis is that RNF213 belongs to the dynein ATPase family, encoding six AAA units in a single polypeptide. Structural alignments

indicated that RNF213 is most similar to the resting states of dynein, in which the AAA ring is also open (*Figure 2—figure supplement 1*). Moreover, RNF213 features two highly conserved loops in the AAA ring, presumably controlling the ATPase activity of the two functional AAA3 and AAA4 units. A striking difference to dynein is the lack of the highly active ATPase AAA1, powering movement along microtubule tracks. We thus propose that RNF213 does not represent a processive ATPase, as would be required for example for it to use its AAA ring to promote the remodeling of E3 target proteins. The distribution of active/inactive ATPase sites rather point to a regulated switch promoting conformational changes by opening and closing the AAA ring, of which the first state is depicted in the present study. Interestingly, the ribosome maturation factor Rea1 – another six AAA-domain containing protein with structural homology to RNF213 (*Figure 2—figure supplement 1*) – was also observed in an auto-inhibited form that is stabilized by a specific insertion into AAA2 (*Chen et al., 2018*; *Sosnowski et al., 2018*). In Rea1, certain cofactors promote the release of the inhibitory element, unlocking the AAA hexamer and inducing conformational changes required to interact with the C-terminal MIDAS domain, and subsequently with other ribosome assembly factors. Given the similar overall organization of Rea1 and RNF213 (*Figure 2—figure supplement 3*), the AAA switch proposed for RNF213 could be analogously transmitted to the C-terminal E3 module, regulating ubiquitination activity.

To characterize the E3 function of RNF213, we used auto-ubiquitination as a proxy readout for its catalytic activity. Our data demonstrate that RNF213 collaborates with UbcH7, a cysteine-reactive E2 enzyme typically stimulating ubiquitination activity of HECT and RBR E3 ligases. Moreover, we observed that RNF213 does not depend on its naming RING domain to function as an E3 ubiquitin ligase. These data are consistent with a recent proteomic study, where a UbcH7-derived activity-based probe (ABP) was developed, lacking those E2 residues required to interact with the E3 RING domain. Contrary to RBR enzymes, the modified UbcH7 ABP exhibited the same cross-linking activity with RNF213 as the unmodified ABP (*Pao et al., 2018*). These in vivo data are consistent with our in vitro findings, demonstrating that the RING is not involved in promoting the transthiolation reaction. Strikingly, the RING-less RNF213, which exhibits strong auto-ubiquitination activity, does not display structural similarity to any of the known ubiquitination enzyme in eukaryotes nor to the novel E3-like ligases (NEL, *Maculins et al., 2016*) that are found in bacterial pathogens and also operate in a RING-independent manner. We thus propose that RNF213 is the founding member of a distinct class of E3 ligases, employing a Cys-containing motif similarly as HECT, RBR, and RCR enzymes, but utilizing a distinct structural scaffold to promote the ubiquitin transfer. Although the E3 mechanism, the identity of the active site cysteine and the role of the RING domain remain to be resolved, our findings highlight the unexplored mechanistic diversity within the ubiquitination network. Remarkably, the majority of the MMD mutations with a high pathology CADD score cluster in the composite E3 module, affecting each of the four sub-domains. These data suggest that MMD mutations interfere with the ubiquitination activity of RNF213, leading to either mechanistic inhibition or altered substrate binding. The presented cryo-EM structure now provides a precise roadmap to dissect the role of RNF213 in vascular formation and the connected Moyamoya disease.

## Materials and methods

**Key resources table**

| Reagent type (species) or resource | Designation | Source or reference | Identifiers | Additional information |
|---|---|---|---|---|
| Gene (*Mus musculus*) | RNF213 | RefSeq | HGNC:14539; RefSeq: NP_ 001035094.2 | |
| Strain, strain background (*Escherichia coli*) | DH10EmBacY | Geneva Biotech | DH10EmBacY | Strain propagated locally from source commercial stock |

*Continued on next page*

*Continued*

| Reagent type (species) or resource | Designation | Source or reference | Identifiers | Additional information |
|---|---|---|---|---|
| Strain, strain background (*Escherichia coli*) | BL21(DE3) | Sigma-Aldrich | CMC0016; RRID:NCBITaxon_469008 | Strain propagated locally from source commercial stock |
| Strain, strain background (AcMNPV) | EmBacY | Geneva Biotech | | |
| Cell line (Spodoptera frugiperda) | Sf9 | Thermo Fisher Scientific | IPLB-Sf-21-AE; RRID:CVCL_0549 | Strain propagated locally from source commercial stock |
| Cell line (Trichoplusia ni) | High Five | Thermo Fisher Scientific | BTI-TN-5B1-4; RRID:CVCL_C190 | Strain propagated locally from source commercial stock |
| Recombinant DNA reagent | pCLA00001 | This paper | addgene: 138607; RRID:Addgene_138607 | Bac-to-Bac compatible plasmid with wt RNF213 |
| Recombinant DNA reagent | pCLA00002 | This paper | addgene: 138608; RRID:Addgene_138608 | Bac-to-Bac compatible plasmid with MMD RNF213 |
| Recombinant DNA reagent | pCLA00003 | This paper | addgene: 138609; RRID:Addgene_138609 | Bac-to-Bac compatible plasmid with ΔRING RNF213 |
| Peptide, recombinant protein | RNF213 wild type | This paper | NP_001035094.2: p. = | Internal ID:Mys1b |
| Peptide, recombinant protein | RNF213 MMD | This paper | NP_001035094.2: p.R4753K | Internal ID:Mys18a |
| Peptide, recombinant protein | RNF213 ΔRING | This paper | NP_001035094.2: p.3941_3999 delinsGSGSG | Internal ID: Mys_Mm_20a |
| Peptide, recombinant protein | Ub_fluor | This paper | NP_066289.3 (1–76): p.1_2insGPLCGS | Cysteine-labelable ubiquitin, based on DOI: 10.1038/s41589-019-0426-z |
| Commercial assay or kit | DSS-H12/D12 | Creative Molecules | 001S | Cross-linking reagent used for XL-MS |
| Commercial assay or kit | DyLight 800 Maleimide | Thermo Fisher Scientific | 46621 | |
| Other | 4–20% Criterion TGX Stain-Free Protein Gel | Bio-Rad | 5678095 | For ubiqutination assay analysis |
| Other | Quantifoil R2/2 Cu 200 | Quantifoil | Quantifoil R2/2 Cu 200 | EM grid used for wt RNF213 |
| Other | UltrAuFoil R2/2 Au 200 | Quantifoil | UltrAuFoil R2/2 Au 200 | EM grid used for MMD RNF213 |
| Software, algorithm | FoldIndex | DOI: 10.1093/bioinformatics/bti537 | RRID:SCR_018390 | |
| Software, algorithm | CADD | DOI: 10.1093/nar/gky1016 | RRID:SCR_018393 | https://cadd.gs.washington.edu/snv |
| Software, algorithm | hhpred | DOI: 10.1093/nar/gki408 | RRID:SCR_010276 | https://toolkit.tuebingen.mpg.de/tools/hhpred |

*Continued on next page*

*Continued*

| Reagent type (species) or resource | Designation | Source or reference | Identifiers | Additional information |
|---|---|---|---|---|
| Software, algorithm | MUSCLE | DOI: 10.1093/nar/gkh340 | RRID:SCR_011812 | https://www.drive5.com/muscle/ |
| Software, algorithm | Fiji | other | RRID:SCR_002285 | https://fiji.sc/ |
| Software, algorithm | Sphire | other | RRID:SCR_018391 | http://sphire.mpg.de/ |
| Software, algorithm | gctf | DOI: 10.1016/j.jsb.2015.11.003 | RRID:SCR_016500 | https://www.mrc-lmb.cam.ac.uk/kzhang/ |
| Software, algorithm | ctffind | DOI: 10.1016/j.jsb.2015.08.008 | RRID:SCR_016732 | https://grigoriefflab.umassmed.edu/ |
| Software, algorithm | Relion | DOI: 10.7554/eLife.42166 | RRID:SCR_016274 | https://www3.mrc-lmb.cam.ac.uk/relion/ |
| Software, algorithm | cryoSPARC | DOI: 10.1038/nmeth.4169 | RRID:SCR_016501 | https://cryosparc.com/ |
| Software, algorithm | UCSF Chimera | DOI: 10.1016/j.jsb.2006.06.010 | RRID:SCR_004097 | https://www.cgl.ucsf.edu/chimera/ |
| Software, algorithm | MotionCor2 | DOI: 10.1038/nmeth.4193 | RRID:SCR_016499 | https://emcore.ucsf.edu/ucsf-motioncor2 |
| Software, algorithm | cryolo | DOI: 10.1038/s42003-019-0437-z | RRID:SCR_018392 | https://sphire.mpg.de/wiki/ |
| Software, algorithm | O | DOI: 10.1107/s0108767390010224 | RRID:SCR_018394 | http://xray.bmc.uu.se/~alwyn/TAJ/Home.html |
| Software, algorithm | USCF ChimeraX | doi: 10.1002/pro.3235 | RRID:SCR_015872 | https://www.rbvi.ucsf.edu/chimerax/ |
| Software, algorithm | Pymol | other | RRID:SCR_000305 | https://pymol.org/2/ |
| Software, algorithm | Phenix | DOI: 10.1107/S2059798318006551 | RRID:SCR_014224 | https://www.phenix-online.org/ |
| Software, algorithm | xiSEARCH | DOI: 10.1074/mcp.M115.049296 | RRID:SCR_018395 | https://www.rappsilberlab.org/software/xisearch/ |
| Software, algorithm | MS amanda | DOI: 10.1021/pr500202e | RRID:SCR_018396 | http://ms.imp.ac.at/?goto=msamanda |
| Software, algorithm | IMP-apQuant | DOI: 10.1021/acs.jproteome.8b00113 | RRID:SCR_018397 | http://ms.imp.ac.at/index.php?action=apQuant |

## Protein sequence analysis

To compare RNF213 protein sequences, the sequence of the mouse orthologue was searched against the NCBI *nr* protein database using NCBI BLAST, and the results manually curated to remove redundant entries and false positives. A multiple sequence alignment was generated with MUSCLE (*Edgar, 2004*), and conservation scores generated with Jalview (*Waterhouse et al., 2009*). The profile shown in *Figure 1a* represents a smoothed conservation plot, applying a sliding window of 100 residues. In *Figure 1—figure supplement 1*, sequence conservation values for each single residue were calculated with the program al2co, using the sum-of-pairs calculation method, no sequence

weight adjustment, and BLOSUM62 as the scoring matrix (*Katoh and Toh, 2008*; *Pei and Grishin, 2001*).

Structurally ordered and disordered regions (*Figure 1—figure supplement 2a*) were predicted with FoldIndex (*Prilusky et al., 2005*). For this purpose, we prepared a composite FoldIndex profile, aligning multiple profiles of the same sequence generated with different window sizes. Each FoldIndex profile was normalized using the reciprocal of the window size and plotted with the areas under the positive (ordered) and negative (disordered) parts of the graph colored green and red, respectively.

Protein domain predictions were carried out with HHpred (*Söding et al., 2005*), initially covering the entire sequence, later the selected sub-regions to confirm the results. Only a few domains were identified with a high confidence score. The identified structural fragments, accounting for less than 15% of the residues altogether, comprised three AAA domains (AAA2, AAA3, AAA4 using the nomenclature in *Figure 2a*) and the RING motif (3940–3999). In the later stages of the structural analysis, the respective RNF213 homology models guided model building into the cryo-EM maps.

## Assembly of RNF213 expression constructs

To obtain recombinant RNF213, we synthesized codon-optimized DNA fragments for insect cell expression. We focused our analysis on the physiologically most relevant mouse RNF213 isoform (RefSeq #NP_001035094.2), adding a $Gly_3$-$His_{10}$ tag to the C-terminus. The 7 cDNA fragments obtained as GeneArt Strings (Thermo Fisher Scientific) were assembled via Gibson Assembly into the pFastBAC1 vector. The final expression vector was confirmed to contain the target sequence by Sanger sequencing. Mutants were generated by splitting the full-length DNA into the separate cloning vectors, performing site-directed mutagenesis, and reassembling the expression vector by Golden Gate Assembly. The mutated cloning sites of the final vectors were validated by Sanger sequencing.

## Expression of RNF213 in insect cells

Source plasmids containing the target constructs were transformed into DH10EMBacY cells. Blue-white screening was used to isolate colonies containing recombinant baculoviral shuttle vectors (bacmids) and bacmid DNA was extracted by alkaline lysis and isopropanol precipitation. Bacmids were then transfected into adherent Sf9 insect cells in 6-well plates, using either Fugene HD transfection reagent (Promega #E2311) or PEI (Polysciences #23966). Successful transfection was tracked by monitoring fluorescence of YFP, encoded by the bacmid backbone. High-titer baculoviral stocks were prepared by transfecting Sf9 suspension cultures. Recombinant protein was expressed at 21°C in High Five insect cells (Thermo Fisher). Cells were harvested 4 days after transfection, and pellets flash-frozen in liquid nitrogen and stored at –80°C. All insect cell culture works were performed at 27°C, using ESF921 serum-free growth medium (Expression Systems #96-001-01) without antibiotic supplementation.

## Purification of RNF213

Frozen cell pellet from 1 L of expression culture was rapidly thawed and resuspended in 50 mL of lysis buffer (50 mM HEPES, 200 mM KCl, 1 mM TCEP, pH 7.2) supplemented with two tablets of Complete EDTA-free Protease Inhibitor (Roche #05056489001) and 200 µL Benzonase (IMP Molecular Biology Service). The lysate was centrifuged for 30 min at $40000 \times g$ to obtain the soluble protein extract. To precipitate DNA, PEI (0.1% w/v final concentration) was added and the suspension incubated for 10 min. After centrifugation, the protein sample was supplemented with 20 mM imidazole and applied to a 5 mL HisTrap FF column (GE Healthcare #17531901) pre-equilibrated with buffer A (50 mM HEPES, 200 mM KCl, 0.25 mM TCEP, 20 mM imidazole, pH 7.2). The column was washed for 30 column volumes with buffer A, after which RNF213 was eluted in a single step using buffer B (50 mM HEPES, 200 mM KCl, 0.25 mM TCEP, 500 mM imidazole, pH 7.2). Fractions containing RNF213 were identified by SDS-PAGE and diluted with buffer C (50 mM HEPES, 0.25 mM TCEP, pH 7.2) to a final conductivity of ~15 mS/cm. After centrifugation for 30 min at $40,000 \times g$, the supernatant was loaded onto a 6 mL Resource Q anion exchange column (GE Healthcare #17117901). The protein was eluted with a linear 0–50% gradient over 20 column volumes against buffer D (50 mM HEPES, 1000 mM KCl, 0.25 mM TCEP, pH 7.2). The pooled RNF213 fractions were

concentrated to a final volume of ~200 μL by Vivaspin ultrafiltration (100 kDa cutoff, Sigma-Aldrich-Sigma-Aldrich # Z614661) and finally applied to a Superose 6 Increase 10/300 GL column (GE Healthcare #29-0915-96) equilibrated with buffer E (50 mM HEPES, 200 mM KCl, 0.25 mM TCEP, pH 7.2). Purified RNF213 was concentrated to >1 mg/mL, flash-frozen in liquid nitrogen and stored at –80°C. All purification steps were carried out at 4°C, using ÄKTA FPLC or ÄKTA pure 25 (GE Healthcare) instruments.

## Preparation of fluorescently labeled ubiquitin

Recombinant human ubiquitin with an N-terminal Met-Gly-Pro-Leu-Cys-Gly-Ser overhang was expressed at 37°C in BL21(DE3) cells using autoinduction medium. Following an established ubiquitin purification procedure (*Ecker et al., 1987*), the expression pellet was resuspended in 50 mM ammonium acetate pH 4.5 and the cells opened by sonication (Branson Digital Sonifier 450, Marshall Scientific). The lysate was centrifuged for 30 min at 40,000 × g, after which the supernatant was heat-denatured for 10 min at 70°C and re-centrifuged. Ubiquitin was finally purified by IEX using SP Sepharose Fast Flow (GE Healthcare) and size exclusion chromatography (SEC) using a Superdex 75 Increase column (GE Healthcare). The protein was concentrated by Vivaspin ultrafiltration (5 kDa cut-off, Sigma-Aldrich #Z614580) to 10 mg/mL. Purified ubiquitin was fluorescently labeled with DyLight800-Maleimide (Thermo Fisher Scientific #46621) and re-purified by IEX as before, and SEC using 5 mM phosphate buffer, pH 7.5. The final ubiquitin sample was concentrated to 5 mg/mL, flash-frozen in liquid nitrogen, and stored at –80°C.

## RNF213 auto-ubiquitination assay

In vitro auto-ubiquitination assays were performed in 10 μL reactions, using 25 mM HEPES, 150 mM NaCl, 10 mM MgCl$_2$, and 2 mM TCEP (VWR #97064–848), pH 8.0 as ligase buffer. The reactions contained 2 mM ATP (Sigma-Aldrich #A2383-5G), 40 μM BSA monomer (Sigma Aldrich #A1900), 0.25 μM human UBE1, 4 μM E2, 20 μM DyLight800-Maleimide-Ubiquitin (as described above), and 0.2 μM RNF213. Recombinant human UBE1 and UbcH7 were provided by the Vienna BioCenter Core Facilities and the Ikeda lab at IMBA, Vienna, respectively. Prior to the E3 ligase activity analysis, the cognate E2, UbcH7, was identified by using a commercially-available kit (E2scan, Ubiquigent #67-0005-001). Reactions were pre-mixed on ice, incubated 30–60 min at 37°C, and stopped by adding SDS-PAGE sample buffer. Proteins were electrophoretically separated on BioRad Stain-Free TGX 4–20% gels. Ubiquitin bands and total protein were visualized by DyLight800 fluorescence and Stain-Free Tryptophan fluorescence, respectively. The images were quantified using Fiji software.

## NADH-coupled ATPase assay

The AAA activity of RNF213 variants was measured in an NADH-coupled ATPase assay using 20 mM HEPES, 200 mM KCl, 2 mM MgCl$_2$, and 0.25 mM TCEP (VWR #97064–848), pH 7.2 as reaction buffer. The final concentrations of the added pyruvate kinase (Sigma-Aldrich #P9136-25KU), lactate dehydrogenase (Sigma-Aldrich #L1254-25KU), phosphoenolpyruvate (Sigma-Aldrich #860077–1G), and NADH (Sigma-Aldrich #N6005-1G) were 5 U/mL, 5 U/mL, 500 μM, and 50 μM, respectively. ATP (Sigma-Aldrich #A2383-5G) was present at the final concentration of 2 mM, with equimolar MgCl$_2$ supplementation. The concentration of RNF213 in the assay was 0.1 μM. A 4 × master mix was prepared with all the components apart from RNF213 and ATP. 2.5 μL of the RNF213 protein sample pre-diluted to 0.4 μM was mixed with 2.5 μL of the master mix and 5 μL of an appropriate pre-diluted ATP stock solution in the same buffer with matched pH. The reactions were pipetted into a 1536-well plate (Greiner Bio-One #782900) and monitored by measuring NADH fluorescence signal over time in a PHERAstar FS plate-reader instrument (BMG Labtech). A 0.05–100 μM NADH dilution series was used as the calibration standard. Reactions were carried out for 12 hr at 30°C. For each curve, the reaction rate was derived from the slope of the region where the signal decayed linearly. By applying the pseudo-zero-order kinetics approximation, ATPase rates were calculated as hydrolyzed ATP molecules per enzyme molecule per minute. Human cytoplasmic dynein (Cytoskeleton #CS-DN01-A) and the actinobacterial Mpa unfoldase were used as a reference for comparison.

### Negative staining EM

The samples were diluted to 0.05–0.1 mg/mL and applied onto carbon-coated Cu/Pd Hexagonal 400 mesh grids (Agar Scientific, #G2440PD). Prior to application of the sample, the grids were glow-discharged on a glass plate with a current of 20 mA for 60 s in the SCD 005 Sputter Coater (BAL-TEC) to clean and hydrophilize the surface. The grids were screened to assess sample quality on a FEI Morgagni microscope equipped with a Morada camera (Olympus-SIS) or a FEI Tecnai G2 20 microscope equipped with a 4 k Eagle camera (FEI). Additionally, promising negative stain grids were imaged on the FEI Tecnai G2 20 using SerialEM (*Schorb et al., 2019*) with a defocus range of –1.5 µm to –2.5 µm and a pixel size of 1.85 Å/px. Negative staining micrographs were analyzed using Relion 2.1 (*Scheres, 2012*). No CTF correction was applied to the images. Around 5000 individual particles were manually picked, from which 2D class averages were generated and used for auto-picking the entire dataset. This dataset was again subjected to 2D classification to assess the conformational variability of RNF213 particles.

### Cryo-EM sample preparation

Frozen aliquots of RNF213 were thawed on ice and diluted in the working buffer (20 mM HEPES, 200 mM KCl, 0.25 mM TCEP, pH 7.2) shortly before freezing cryo-EM grids. Cryo-EM grids were prepared using the EM GP freeze plunger (Leica Microsystems) using Whatman Filter Paper Grade 1 (GE Healthcare #1001–055) as the blotting substrate and a relative humidity of 80–90% and a temperature of 4°C in the sample grid chamber. As cryogenic agent, liquid ethane (Ethane N45, Air Liquide #P0502M20R0A001) adjusted to –182°C was used. Grids were clipped into Titan-Krios-compatible autoloader cartridges ~ 24 hr after preparation and stored as such until screening and data collection.

For wt RNF213, R2/2 Cu 200 mesh grids (Quantifoil) pre-floated onto custom-made 2.9 nm continuous carbon film were used. The grids were placed on a glass slide and glow-discharged with a current of 20 mA for 60 s in a SCD 005 Sputter Coater (BAL-TEC). The wt protein was diluted to 0.05 mg/mL and 5 µL were applied to the grids. The final grid was blotted for 2 s with the Z-position, H position, and post-sensor offset set to 3.5, 178, and +8, respectively.

For R4753K RNF213, UltrAuFoil R2/2 Au 200 mesh grids (Quantifoil) without an additional continuous film were used. These grids were placed on a metallic mesh carrier and glow-discharged with a current of 20 mA for 60 s. The MMD variant protein was diluted to 0.4 mg/mL and 3.5 µL were applied to the grids. The final grid was blotted for 1.5 s with the Z-position, H position, and post-sensor offset set to 3.0, 182, and +6, respectively.

### Screening of cryo-EM grids and data collection

Cryo-EM grids were screened in-house using either a FEI Polara equipped with a K2 camera (Gatan), or a Glacios equipped with a Falcon 3 camera (Thermo Fisher Scientific). Grids of wt RNF213 were further screened at EMBL Heidelberg. Selected grids for both constructs were recorded on a Titan Krios equipped with a K2 camera at the EMBL Heidelberg. Further acquisition parameters are listed in *Supplementary file 3*.

### Image processing and EM map reconstruction

The image processing was carried out predominantly in relion 3.0 (*Zivanov et al., 2018*), with other software being used for smaller subroutines in the process.

For the wt protein, motion correction of the multi-frame micrographs was done with Motioncor2 1.0.5 (*Zheng et al., 2017*) and CTF parameters were estimated using gctf 1.06 (*Zhang, 2016*). From a random subset of 10 micrographs, all visible particles were selected, yielding ~850 particles. A neural network model was trained using cryolo 1.1.1 (*Wagner et al., 2019*) using YOLO architecture with $3 \times 3$ patches per micrograph. The model was used to pick particles from all micrographs, selecting a threshold of 0.1 that yielded maximal number of particles while allowing only a small amount of false positive hits. For cryolo training and picking, a gaussian lowpass filter at 20 Å was found to be optimal. Particle coordinates were imported into Relion and subjected to multiple rounds of 2D classification, discarding classes showing obvious artifacts. Particles from clean classes were exported to Cryosparc v0 (*Punjani et al., 2017*), and an initial reference model was generated and refined. The refined map generated by Cryosparc was imported back into Relion and subjected

to one round of 3D classification, after which refinement yielded a 3.4 Å map. This map was low-pass filtered and projected for a new round of auto-picking yielding extra particles representing projections that were underrepresented during initial picking. These particles were analogously processed as the cryolo-picked ones. Duplicate particles were excluded by enforcing a minimal interparticle distance of 50 Å, after which the particle sets were combined, and further refined by Bayesian Polishing and CTF Refinement, yielding a 3.2 Å overall map with a non-uniform local resolution (*Figure 1—figure supplement 2*). Following that, rigid body boundaries within the structure were identified by performing 3D refinements using masks constrained to different parts of the map and observing which regions remain well-refined even when masked-out (i.e. co-refine). Soft masks were generated using Sphire for the three identified rigid bodies (N-arm, AAA, E3 – see *Figure 1—figure supplement 2*). The partitioning was repeated yielding a final set of four rigid body zones (N-arm, AAA, E3a, E3b). Focused Refinements were carried out for all zones, yielding maps with improved resolution. A composite map from all focused maps was calculated in UCSF Chimera (*Goddard et al., 2007*).

For the MMD variant, motion correction was done using the native Relion implementation of the algorithm and CTF parameters were estimated using CTFFIND4 (*Rohou and Grigorieff, 2015*). The cryolo model trained on wt protein was used to auto-pick particles from the full set of micrographs with a 20 Å lowpass filter and an optimized picking threshold. Separately, the auto-picking was performed with Relion, where the picking reference was the wt EM density map lowpass-filtered to 10 Å. The two coordinate sets were processed separately in Relion 3.0 analogously to the wt RNF213 map, using a relion-generated ab initio model of the MMD variant as a reference. The particle sets were combined and further refined as for the wt map, applying the same masks for the Focused Refinement as for the wt map.

## Building the molecular model of RNF213

Owing to the slightly better resolved cryo-EM density of the R4753K mutant, as compared to the wt protein, we used the respective focused EM maps for chain tracing of RNF213. The high resolution of these maps (N-arm: 3.4 Å; AAA: 3.0 Å; E3: 3.1 Å) enabled us to build the entire model de novo (residues 475–5148; except flexible regions indicated in *Supplementary file 1*). It was even possible to include side chains for >80% of the residues with high accuracy. Model building was carried out with the program O (*Jones et al., 1991*), using its Lego_Ca command to build non-regular loop regions of the molecule. Throughout the model building progress, we used XL-MS data, resulting from triplicate measurements of both wt and RNF213 samples, to carefully cross-check the inserted and assigned protein portions. In the final model, 90% of the DSS-mediated crosslinks observed between lysine residues are separated by 15–30 Å (*Figure 2—figure supplement 1*). We also confirmed the fold of AAA2/3/4 and the RING with HHpred homology models, derived from apo dynein (4aki, *Schmidt et al., 2012*) and TRIM25 (5eya, *Sanchez et al., 2016*). To further minimize model bias, we carried out simulated annealing, implemented in the real-space refinement procedure of Phenix (*Afonine et al., 2018*). Afterwards, the refined wt and R4753K structures were checked for proper stereochemistry by EM-RINGER (*Barad et al., 2015*; *Table 1*). Owing to the slightly better quality of the R4753K density and the derived model, all structural illustrations were made with this variant, using either PyMOL (*Schrodinger LLC, 2015*) or UCSF Chimera (*Goddard et al., 2007*).

## Cross-linking mass spectrometry (XL-MS)

For XL-MS experiments 0.7 mg/mL of purified RNF213 supplied in 150 mM NaCl, 20 mM HEPES pH 7.5 was crosslinked with 0.35 mM DSS (Creative Molecule) for 40 min at 25°C. The reaction was quenched with 50 mM ammonium bicarbonate (Sigma) for 10 min at 37°C. Samples were dried at 45°C (Concentrator, Eppendorf), resuspended in 8 M Urea (VWR) and reduced with 2.5 mM TCEP (VWR). Subsequently, samples were alkylated with 5 mM iodoacetamide (Sigma) for 30 min at room temperature in the dark. Urea was then diluted to 1 M by adding 50 mM ammonium bicarbonate. The samples were digested with trypsin (Promega), using 2 μg of protease per 100 μg protein. After 20 hr at 37°C, trypsin was inactivated by adding 0.4% (v/v) trifluoroacetic acid (Thermo Fisher). Samples were loaded on Sep-Pak cartridges (50 mg, Waters) equilibrated with 5% (v/v) acetonitrile (VWR), 0.1% (v/v) formic acid (Fisher Chemicals) and subsequently eluted with 50% (v/v) acetonitrile, 0.1% (v/v) formic acid. To enrich for crosslinked peptides, the samples were separated by SEC. For

this purpose, peptides were resuspended in 30% (v/v) acetonitrile, 0.1% (v/v) trifluoroacetic acid and applied to a Superdex 30 Increase 3.2/300 column (GE Healthcare). Fractions containing crosslinked peptides were evaporated to dryness.

For MS-analysis samples were resuspended in 5% (v/v) acetonitrile, 0.1% (v/v) TFA. The nano HPLC system used was an UltiMate 3000 RSLC nano system (Thermo Fisher Scientific) coupled to an Orbitrap Fusion Lumos Tribrid mass spectrometer (Thermo Fisher Scientific), equipped with a Proxeon nanospray source (Thermo Fisher Scientific). Peptides were loaded onto a trap column (Thermo Fisher Scientific, No. 160454) at a flow rate of 25 µL/min using 0.1% TFA as mobile phase. After 10 min, the trap column was switched in line with the analytical column (Thermo Fisher Scientific, No. 164739). Peptides were eluted using a flow rate of 230 nL/min, and a binary 3 hr gradient. The gradient starts with 98% mobile phase A (water/formic acid, 99.9/0.1, v/v) and 2% mobile phase B (water/acetonitrile/formic acid, 19.92/80/0.08, v/v/v), increases to 35% mobile phase B over 180 min, followed by a 5 min-gradient to 90% mobile phase B. Acquisition was performed in data-dependent mode with a 3 s cycle time. The full scan spectrum was recorded at a resolution of 60 000 in the range of 350–1500 m/z. Precursors with a charge state of +3 to +7 were fragmented. HCD-collision energy was set to 29%. The resolution of MS2-scans recorded in the Orbitrap was 45 000 with a precursor isolation width of 1.0 m/z. Dynamic exclusion was enabled with 30 s exclusion time.

Fragment spectra peak lists were generated from the raw MS-data using the software MSConvert (v 3.0.9974) (*Chambers et al., 2012*) selecting the peak picking filter. Crosslink search was performed using the XiSearch (v 1.6.742) (*Giese et al., 2016*) applying the following parameters: 6 ppm $MS^1$-accuracy; 20 ppm $MS^2$-accuracy; DSS-crosslinker with reaction specificity for lysine, serine, threonine, tyrosine, and protein N-termini with a penalty of 0.2 (scale 0–1) assigned for serine, threonine, and tyrosine; carbamidomethylation of cysteine as a fixed modification; oxidation of methionine as variable modification; tryptic digest with up to four missed cleavages; and all other variables at default settings. Identified crosslinks were filtered to 5% FDR on link level with the software XiFDR (v 1.1.27) (*Fischer and Rappsilber, 2017*; *Supplementary file 4*). For analysis of the crosslinked data, the in-house software CrossLinkingVisualizer has been used (*Ahel, 2020*).

## Mapping of RNF213 mutation severity scores

A comprehensive list of RNF213 mutations implicated with MMD was extracted from a recent publication (*Gagunashvili et al., 2019*). The list was pre-curated, including only missense point mutations and removing the annotated mutations that did not fit the reference RNF213 sequence (Uniprot #Q63HN8-3). All mutations were mapped to genomic coordinates in the GRCh38 human genome assembly, after which a Combined Annotation Dependent Depletion (CADD) score was assigned for each using CADD v1.5 (*Rentzsch et al., 2019*; *Supplementary file 2*). The CADD score was used as a proxy for the pathologic severity of each mutation and was finally mapped onto the RNF213 structure (*Figure 4c*). Residues numbers were transferred from the human to the mouse protein according to the sequence alignment shown in *Supplementary file 1*.

## Data availability

Atomic coordinates and cryo-EM density maps have been deposited in the Protein Data Bank (PDBe) under accession codes EMD-10429 resp. 6TAX for the wildtype and EMD-10430 resp. 6TAY for the MMD variant. The raw micrographs were submitted to the EMPIAR database (EMPIAR-10334). The mass spectrometry proteomics data have been deposited to the ProteomeXchange Consortium via the PRIDE (*Perez-Riverol et al., 2019*) partner repository with the dataset identifier PXD018701. All other source data are included in the paper.

## Acknowledgements

We thank all members of the Clausen group and M Suskiewicz (Oxford University) for remarks on the manuscript and discussions, and Lilian Fennell (IMBA, Vienna) for support in setting up ubiquitination assays. Samples were prepared and screened at the EM Facility of the Vienna BioCenter Core Facilities GmbH (VBCF). The cryo-EM data were collected at the cryo-EM platform of the European Molecular Biology Laboratory in Heidelberg, overseen by Felix Weiss. This work was supported by a grant from the European Research Council (AdG 694978, to TC) and by FFG Headquarter Grant 852936 (to TC). The IMP is supported by Boehringer Ingelheim.

## Additional information

### Funding

| Funder | Grant reference number | Author |
|---|---|---|
| European Commission | ERC AdG 694978 | Tim Clausen |

The funders had no role in study design, data collection and interpretation, or the decision to submit the work for publication.

### Author contributions

Tim Clausen, Conceptualization, Supervision, Funding acquisition, Validation, Investigation, Visualization, Methodology, Writing - original draft, Project administration, Writing - review and editing; Juraj Ahel, Data curation, Software, Formal analysis, Validation, Investigation, Visualization, Methodology, Project administration, Writing - review and editing; Anita Lehner, Antonia Vogel, Investigation, Visualization, Writing - review and editing; Alexander Schleiffer, Anton Meinhart, Investigation, Methodology, Writing - review and editing; David Haselbach, Software, Supervision, Validation, Investigation, Visualization, Methodology, Project administration, Writing - review and editing

### Author ORCIDs

Juraj Ahel (iD) https://orcid.org/0000-0003-0293-1063
David Haselbach (iD) http://orcid.org/0000-0002-5276-5633
Tim Clausen (iD) https://orcid.org/0000-0003-1582-6924

### Decision letter and Author response

Decision letter https://doi.org/10.7554/eLife.56185.sa1
Author response https://doi.org/10.7554/eLife.56185.sa2

## Additional files

### Supplementary files

• Supplementary file 1. Multiple alignment of RNF213 orthologs. Sequences were retrieved from NCBI non redundant protein database or from UniProt reference proteomes with the following accessions: *Mus musculus* (NCBI: ref|NP_001035094.2), *Homo sapiens* (Uniprot: sp|Q63HN8), *Gallus gallus* (NCBI: ref|XP_015151083.1), *Xenopus laevis* (Uniprot: tr|A0A1L8ETH7), Danio rerio (Uniprot: sp|A0A0R4IBK5); sequences were aligned with MAFFT version 7.427 (27), and visualized with Jalview[26]. Alpha helices (grey) and beta strands (black) are derived from the cryo-EM structure and shown on top. Residues with no structural data are indicated by a dashed line. At the bottom, positions with MMD mutations are marked by polygons, where green represents a low (<15), magenta an intermediate (<20) and red a high CADD score (>=20). Arrows indicate domain borders. Within the AAA+ domains, functional residues for nucleotide binding and hydrolysis are indicated by letters (A, Walker A; B, Walker B; S1, sensor I; S2, sensor II; RF, arginine finger). Regions IR3, IR5, and the E3-RING are enframed.

• Supplementary file 2. RNF213 variants associated with MMD. For each variant, listed are the corresponding residue in mouse RNF213, whether the residue is conserved between mouse and human, the cDNA variants considered, and the CADD score for the least-scoring cDNA variant. The domain each residue belongs to is also indicated. CADD scores >= 20 are highlighted in red. Variant annotations refer to the RNF213 isoform 1 (RefSeq NM_001256071.3, human genome assembly GRCh38. p13). The lower panel shows the distribution per RNF213 module, highlighting the accumulation of MMD mutations in the composite E3 domain.

• Supplementary file 3. Imaging and reconstruction details of wt and R4753K RNF213.

• Supplementary file 4. Processed crosslinking MS data of wt RNF213.

• Transparent reporting form

## Data availability

Atomic coordinates and cryo-EM density maps have been deposited in the Protein Data Bank (PDBe) under accession codes EMD-10429 resp. 6TAX for the wildtype and EMD-10430 resp. 6TAY for the MMD variant. The raw micrographs were submitted to the EMPIAR database (EMPIAR-10334). The mass spectrometry proteomics data have been deposited to the ProteomeXchange Consortium via the PRIDE partner repository with the dataset identifier PXD018701. All other source data are included in the paper.

The following datasets were generated:

| Author(s) | Year | Dataset title | Dataset URL | Database and Identifier |
|---|---|---|---|---|
| Ahel J, Meinhart A, Haselbach D, Clausen T | 2020 | Mouse RNF213 wild type protein | http://dx.doi.org/10.2210/pdb6tax/pdb | Worldwide Protein Data Bank, 10.2210/pdb6tax/pdb |
| Ahel J, Meinhart A, Haselbach D, Clausen T | 2020 | Mouse RNF213 mutant R4753K modeling the Moyamoya-disease-related Human variant R4810K | http://doi.org/10.2210/pdb6tay/pdb | Worldwide Protein Data Bank, 10.2210/pdb6tay/pdb |
| Ahel J, Meinhart A, Haselbach D, Clausen T | 2020 | Mouse RNF213 wild type protein | https://www.ebi.ac.uk/pdbe/entry/emdb/EMD-10429 | Electron Microscopy Data Bank, EMD-10429 |
| Ahel J, Meinhart A, Haselbach D, Clausen T | 2020 | Mouse RNF213 mutant R4753K modeling the Moyamoya-disease-related Human variant R4810K | https://www.ebi.ac.uk/pdbe/entry/emdb/EMD-10430 | Electron Microscopy Data Bank, EMD-10430 |
| Ahel J, Lehner A, Vogel A, Schleiffer A, Meinhart A, Clausen T, Haselbach D | 2020 | Cryo-EM structure of RNF213 reveals a RING-type E3 with a dynein core and cysteine reactivity | http://doi.org/10.6019/EMPIAR-10334 | EMPIAR, 10.6019/EMPIAR-10334 |
| Ahel J, Lehner A, Vogel A, Schleiffer A, Meinhart A, Haselbach D, Clausen T | 2020 | Moyamoya disease factor RNF213 is a giant E3 ligase with a dynein-like core and a distinct ubiquitin-transfer mechanism | http://doi.org/10.6019/PXD018701 | PRIDE:PXD018701, 10.6019/PXD018701 |

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
