## [Decision Letter]

**Acceptance summary:**

RNF213 is the major susceptibility gene for Moyamoya disease and is the largest single polypeptide E3 ubiquitin ligase known. The precise function of this giant E3 ligase is not well understood at present though it has been suggested to function as a metabolic gatekeeper. This manuscript describes the cryo-EM structure of mouse RNF213 and provides the first analysis of disease-associated mutations. This is an exciting story that highlights how structural biology can provide insight into the potential function of proteins that lack annotated domains for most of their sequence.

**Decision letter after peer review:**

Thank you for submitting your article "Moyamoya disease factor RNF213 is a giant E3 ligase with a dynein-like core and a distinct ubiquitin-transfer mechanism" for consideration by *eLife*. Your article has been reviewed by three peer reviewers, including Ivan Dikic as the Reviewing Editor and Reviewer #1, and the evaluation has been overseen by David Ron as the Senior Editor The following individual involved in review of your submission has agreed to reveal their identity: NIng Zheng (Reviewer #2).

The reviewers have discussed the reviews with one another, and the Reviewing Editor has drafted this decision to help you prepare a revised submission.

We would like to draw your attention to changes in our revision policy that we have made in response to COVID-19 (https://elifesciences.org/articles/57162). Specifically, we are asking editors to accept without delay manuscripts, like yours, that they judge can stand as *eLife* papers without additional data, even if they feel that they would make the manuscript stronger. Thus, the revisions requested below only address clarity and presentation.

This manuscript describes a remarkable structure of a very large ubiquitin E3 ligase, known as RNF213, which contains more than 5000 amino acids. Mutations of RNF213 are associated with a human cerebrovascular disorder called Moyamoya Disease (MMD). Because of its giant size, biochemical and functional analyses of the protein have been challenging. By leveraging the power of cryo-EM, the authors have determined the structure of RNF213, which reveals a complex architecture highlighted by a dynein-like six AAA ATPase domain core and a novel multi-domain E3 scaffold. Based on structural comparison analysis, the authors show that the AAA ring of RNF213 has a topology similar to the resting states of dynein and features distinct structural elements that argue against a processive ATPase. Perhaps one of the most stunning findings of the study is the RING-independent E3 activity of RNF213, which hints at a new class of ubiquitin E3 ligases. Another mystery arising from the structural studies is the functional role of two interacting charged residues, R4753, and D4806, located in the E3 module, which are mutated in MMD. Structural comparison between the wt and R4753K mutant form of RNF213 reveals only subtle conformational differences.

Although the study does not yet conclusively establish some concrete structure-function relationships of RNF213, it does provide, for the first time, the necessary structural framework for dissecting the multifaceted functions of this enormous E3 protein. There are no serious concerns regarding this manuscript. However, there are a few suggestions, which might add an additional level of novelty and excitement as well as some changes to the presentation that could be made that the work is more easily accessible to the reader.

1) The authors used auto-ubiquitination as a readout for E3 activity. It would be interesting to know if the ubiquitination of one of its substrates involved in MMD and focus on any potential differences between wild type and mutant RNF213. Have the authors done that? It might be possible that auto-ubiquitination is unaffected, but the ubiquitination of its substrate(s) changed.

2) The SEC elution profile shown in Figure 1—figure supplement 1B provides evidence that the sample is a single species but in the absence of a MW calibration curve or SEC-MALLS data does not provide any information about the monomeric/oligomeric state of the protein. Please either show such data if they are available or otherwise rephrase the sentence.

3) The statement in subsection “RNF213 is a distinct E3 ubiquitin ligase” that UbcH7 is among the less reactive E2s is incorrect. UbcH7 is a cysteine-reactive E2. This section needs to be rephrased to make it explicitly clear what the difference between UbcH7 and other, lysine-reactive E2s is. And why the observation that RNF213 works with UbcH7 is so remarkable. Please make sure to include all relevant references to E3s that function via thioester intermediates.

Please also comment in the Results section if there is any structural or sequence homology between the E3 module of RNF213 and those E3s that contain a catalytic cysteine.

Based on this structure, could the authors speculate how RNF213 might work?

---

## [Author Response]

[…]1) The authors used auto-ubiquitination as a readout for E3 activity. It would be interesting to know if the ubiquitination of one of its substrates involved in MMD and focus on any potential differences between wild type and mutant RNF213. Have the authors done that? It might be possible that auto-ubiquitination is unaffected, but the ubiquitination of its substrate(s) changed.

The reviewers raise an important point. It would be very informative to test whether the ability of RNF213 to ubiquitinate its putative substrates is affected by MMD mutations, which do not seem to effect auto-ubiquitination activity. Unfortunately, to date no direct substrates have been identified. From the implicated targets, we have tested NFATC2, but could not detect substantial ubiquitination activity of RNF213. Most likely, the published association between NFATC2 and RNF213 is indirect, relying on other E3 enzymes operating in between.

In terms of RNF213 mechanism, our data demonstrate that the basal E3 activity, i.e. the ability to catalyze the transfer of ubiquitin from an E2 to a protein substrate, is preserved in the ∆RING deletion and the R4753K MMD mutants. We therefore consider it likely that either the RING domain that harbors many of the most severe MMD variants, or other pathologic mutations in the E3 module, affect ubiquitination by disrupting the substrate binding site of RNF213.

Moreover, we noted that the rate of self-ubiquitination is much higher than the rates of ubiquitination of other reaction components (E1, E2, BSA, free ubiquitin), with the majority of ubiquitin being attached to RNF213 when the reaction is running to completion (Author response image1). This apparent specificity for auto-ubiquitination may hint to a physiologically relevant activity, however, needs to be further explored.

**Author response image 1. sa2fig1:** Full gel images used to quantify wt and R4753K RNF213 auto-ubiqitination activity. (a) The ubiquitin signal (via DyLight 800 label, top panel) and the total protein signal (via tryptophan Stain-Free in-gel labeling, bottom panel) are shown separately. Yellow triangles point to ubiquitinated species. The shallow gradient on top indicates the concentration of RF213 used (80%-120% of the nominal value 0.4 µM). (b) Densitometric plot of the positive control, shown in lane 13:(+). Despite RNF213 being a minor protein component in the mix, the majority of ubiquitin is transfered onto RNF213 rather than on more abundant species like BSA or the E2. For the densitometry of the Ub signal, the saturated signal excluding free ubiquitin is shown in grey.

2) The SEC elution profile shown in Figure 1—figure supplement 1B provides evidence that the sample is a single species but in the absence of a MW calibration curve or SEC-MALLS data does not provide any information about the monomeric/oligomeric state of the protein. Please either show such data if they are available or otherwise rephrase the sentence.

We have rephrased the text to clarify how the conclusion was derived: EM studies show homogenous monomeric particles, DLS measurement reveal the size and monodispersity of the analyzed monomer, as further confirmed by a SEC analysis. To clarify this point, we have now added the DLS data and have rephrased the corresponding paragraph in the main text:

“The initial cryo-EM class averages revealed a compact macromolecule with overall dimensions of 90 × 130 × 220Å^3^, depicting the RNF213 ubiquitin ligase in its monomeric state. Size exclusion chromatography (SEC) and dynamic light scattering (DLS) analyses pointed to a monodisperse protein population (Figure 1—figure supplement 1BC), suggesting that the monomer is the dominant form of RNF213 in solution.”

3) The statement in subsection “RNF213 is a distinct E3 ubiquitin ligase” that UbcH7 is among the less reactive E2s is incorrect. UbcH7 is a cysteine-reactive E2. This section needs to be rephrased to make it explicitly clear what the difference between UbcH7 and other, lysine-reactive E2s is. And why the observation that RNF213 works with UbcH7 is so remarkable. Please make sure to include all relevant references to E3s that function via thioester intermediates.Please also comment in the Results section if there is any structural or sequence homology between the E3 module of RNF213 and those E3s that contain a catalytic cysteine.Based on this structure, could the authors speculate how RNF213 might work?

We thank the reviewers for clarifying the point of UbcH7 reactivity. The text has been rephrased accordingly, noting the inability of UbcH7 to discharge onto lysine residues.

“This finding is surprising, because UbcH7 is not known to collaborate with canonical RING-type E3s as it lacks intrinsic reactivity with lysine. Instead, UbcH7 is reactive only with cysteine residues and works together with transthiolation E3 enzymes such as HECT (Schwarz, Rosa and Scheffner, 1998), RBR (Wenzel et al., 2011), and RCR (Pao et al., 2018) ubiquitin ligases.”

Of note, when looking for structurally related domains of the E3 module in the entire PDB database, we did not find a single sub-domain matching more than 3 helices in any target, highlighting the unique nature of the RNF213 ubiquitination scaffold. Likewise, we could not detect sequence similarity to any other E3 enzyme, beyond the RING domain itself and short helical repeats. To account for these comparative analyses, we added a short statement in the Discussion part:

“Strikingly, the RING-less RNF213, which exhibits strong auto-ubiquitination activity, does not display structural similarity to any of the known ubiquitination enzyme in eukaryotes nor to the novel E3-like ligases (NEL, Maculins et al., 2016) that are found in bacterial pathogens and also operate in a RING-independent manner. […] Though the E3 mechanism, the identity of the active site cysteine and the role of the RING domain remain to be resolved, …”

Given the unknown identity of the E3 active site as well as of the nucleophile mediating the ubiquitin transfer from E2 to substrate, we cannot speculate on the molecular mechanism. Collaboration with UbcH7 suggests that the transthiolation reaction proceeds by a Cys-relay ping-pong mechanism. However, this is already stated in the manuscript.